# Coral Gardens Reef, Belize: An *Acropora* spp. refugium under threat in a warming world

Lisa Greer[1]*, H. Allen Curran[2], Karl Wirth[3], Robert Humston[4], Ginny Johnson[1], Lauren McManus[1,5], Candice Stefanic[1,6], Tara Clark[7,8], Halard Lescinsky[9], Kirah Forman-Castillo[10]

**1** Department of Earth and Environmental Geoscience, Washington and Lee University, Lexington, Virginia, United States of America, **2** Department of Geosciences, Smith College, Northampton, Massachusetts, United States of America, **3** Department of Geology, Macalester College, St. Paul, Minnesota, United States of America, **4** Department of Biology, Washington and Lee University, Lexington, Virginia, United States of America, **5** Kogod School of Business, American University, Washington, DC, United States of America, **6** Department of Anatomical Sciences, Stony Brook University, Stony Brook, New York, United States of America, **7** School of Earth, Atmospheric, and Life Sciences, University of Wollongong, Wollongong, New South Wales, Australia, **8** Radiogenic Isotope Facility, School of Earth and Environmental Sciences, The University of Queensland, Brisbane, Queensland, Australia, **9** Department of Biology and Earth Science, Otterbein University, Westerville, Ohio, United States of America, **10** Hol Chan Marine Reserve, San Pedro, Belize

* greerl@wlu.edu

**Data Availability Statement:** All relevant data are within the paper and its Supporting information files.

## Abstract

Live coral cover has declined precipitously on Caribbean reefs in recent decades. *Acropora cervicornis* coral has been particularly decimated, and few Western Atlantic *Acropora* spp. refugia remain. Coral Gardens, Belize, was identified in 2020 as a long-term refugium for this species. This study assesses changes in live *A. cervicornis* coral abundance over time at Coral Gardens to monitor the stability of *A. cervicornis* corals, and to explore potential threats to this important refugium. Live coral cover was documented annually from 2012–2019 along five permanent transects. *In situ* sea-surface temperature data were collected at Coral Gardens throughout the study period and compared with calibrated satellite data to calculate Maximum Monthly Mean (MMM) temperatures and Degree Heating Weeks (DHW). Data on bathymetry, sediment, substrate, herbivore abundance, and macroalgal abundance were collected in 2014 and 2019 to assess potential threats to Coral Gardens. Live coral cover declined at all five transect sites over the study period. The greatest loss of live coral occurred between 2016 and 2017, coincident with the earliest and highest maximum average temperatures recorded at the study site, and the passage of a hurricane in 2016. Structural storm damage was not observed at Coral Gardens, though live coral cover declined after the passage of the storm. Uranium-thorium ($^{230}$Th) dating of 26 dead *in situ* fragments of *A. cervicornis* collected in 2015 from Coral Gardens revealed no correlation between coral mortality and tropical storms and hurricanes in the recent past. Our data suggest that several other common drivers for coral decline (i.e. herbivory, predation, sedimentation, pH) may likely be ruled out for Coral Gardens. At the end of the study period, Coral Gardens satisfied most criteria for refugium status. However, the early onset, higher mean, and longer duration of above-average temperatures, as well as intermittent temperature anomalies likely played a critical role in the stability of this refugium. We suggest that

**Funding:** This work was supported by the National Science Foundation and the Keck Geology Consortium under Grant No. NSF-REU #1358987 to LG, KW and HL and Grant No. NSF-REU #1659322 to LG and KW (https://keckgeology.org/). Support also was received from The Washington and Lee University: Johnson Opportunity Grant, Summer Research Scholar Program, Lenfest summer research grant, R. Preston Hawkins IV Award in Geology, and Department of Geology to LG, GJ, LM, CS, and ARC DECRA Fellowship support for TC (https://www.arc.gov.au/grants/discovery-program/discovery-early-career-researcher-award-decra). The funders had no role in study design, data collection and analysis, decision to publish, or preparation of the manuscript.

**Competing interests:** The authors have declared that no competing interests exist.

temperature stress in 2016 and perhaps 2015 may have increased coral tissue vulnerability at Coral Gardens to a passing hurricane, threatening the status of this unique refugium.

## Introduction and background

*Acropora* corals (*A. cervicornis* and *A. palmata*) have historically provided the structural support for many Caribbean Pleistocene and Holocene reefs, creating complex architectural framework and substrate for shallow-water reef dwellers [1, 2]. The more fragile of the two acroporids, *A. cervicornis* (staghorn coral), is generally found living in shallow waters (5–20 m) on fore- and patch reefs [3] and has a growth rate of between three to over 11 cm per year [4, 5]. The fossil record shows that acroporid corals are a well-documented component of pre-modern reefs in Belize [6–8], and studies suggest that acroporids have persisted as an important component of Caribbean reefs in the face of climate and environmental change in the geologic past [9–13]. Nonetheless, some estimates suggest that acroporids have declined by up to 98% since the 1980s, or a few decades prior [14, 15]. Coral Gardens, Belize has been identified as a refugia for these endangered species [16].

While other extant acroporid-dominant coral communities have been reported in the last decade or so [17–19], few of these are reported to be extensive and/or true reef-building structures [20, 21]. The degree to which any of these sites may also qualify as an *Acropora* spp. refugium remains in question, making Coral Gardens a valuable site for monitoring this species. Coral reef refugia were first recognized as habitats protected from rapid temperature rise and coral bleaching in 1996 [22], but have been defined in numerous ways using differing criteria [23–25]. In general, a refugium is a place where a species may retreat to, persist in, and expand from [24]. Low exposure to stressors can be key to refugium maintenance. Habitats that provide a buffer for organisms from climate stressors on timescales shorter than decades, healthy reefs that are located in remote locations distant from anthropogenic impacts, and reefs primarily composed of species that are tolerant to extreme climatic conditions should not be counted as true refugia according to Kavousi and Keppel [25]). Longevity is key to refugium designation but assessing how long a reef has existed in a given area can be particularly challenging unless it has been monitored for decades.

In 2017, the International Union for Conservation of Nature (IUCN) listed the Belize Barrier Reef system as an area of "significant concern." This report cited habitat shifts, sea-level rise, temperature extremes, and potential increase in hurricanes as a "very high threat," and overfishing, coastal development, tourism, sedimentation, and ocean acidification as a "high threat" to the future integrity of Belize reefs [26]. Contrary to these concerns, living coral communities within Coral Gardens are comprised of up to 75,000 m$^2$ of extensive framework-building acroporid thickets [27]. Irwin et al. [28] used a novel genetic aging technique [29] to show that some genets of *A. cervicornis*, *A. palmata*, and even the F1 hybrid *A. prolifera* are long-lived (up to hundreds of years in age) at Coral Gardens. Reef longevity was assessed using more traditional radiometric dating techniques to show that *A. cervicornis* presence at Coral Gardens at least pre-dated and persisted through the widespread Caribbean acroporid die-offs during the 1980s and earlier [16]. This site therefore provides a unique opportunity to assess and monitor the continued viability of a long-standing endangered coral species refugium.

Numerous local and global factors are known to threaten coral reefs [30–32]. Coral reefs are not uniform in the pressures they face or how they respond to these threats [33]. Rising sea-surface temperatures (SST), particularly in years where El Niño-related warming compounds the effects of global warming, increasingly stress corals and can induce coral bleaching

[34–36]. Excessive macroalgae growth due to overfishing, eutrophication, or loss of key reef herbivores can inhibit coral recruitment and the ability of reefs to recover from disturbance [37, 38]. Tropical storms and hurricanes have always impacted reefs, but these events may be increasing in strength or frequency as a result of a changing climate [39]. Land-based pollution, sedimentation, and contaminants resulting from coastal development and land use can also negatively impact reefs [40]. Unmitigated or poorly managed marine tourism may result in harm to reefs through boat groundings, anchor damage, or snorkeler/diver damage [41], and even when Marine Protected Areas are designated, location, size, lack of connectivity with other safe spaces, or inadequate enforcement can exacerbate problems for coral reef communities at these sites [42]. Excessive bioerosion, predation, or loss of carbonate accretion can have detrimental impacts to the structural integrity of reefs [43]. Finally, one of the most damaging effects on acroporid coral abundance throughout the wider Caribbean is White Band Disease (WBD), which has been devasting *Acropora* spp. corals for at least the last few decades [44, 45].

This study aims to evaluate the potential threats to the Coral Gardens refugium based on available data. In this study, we surveyed live coral abundance at an existing *A. cervicornis* refugium annually from 2012 through summer 2019. Our data show that while this refuge still persists, it remains under threat. Measurements of live coral cover, *in situ* SST data, $^{230}$Th dating of dead coral fragments, macroalgae and herbivore abundance, sediments, and personal observations in the field were used to evaluate potential drivers of declining coral cover at Coral Gardens. Using a heuristic process of elimination, we suggest that while most 'threat variables' did not change at Coral Gardens during the study, thermal stress may have increased the vulnerability of *A. cervicornis* to storms.

## Study area

Coral Gardens Reef (17˚50'00.36"N, 87˚59'32.45"W) is located inshore of the Mesoamerican Barrier Reef between the islands of Ambergris Caye and Caye Caulker. It sits between two marine protected areas, the Hol Chan and the Caye Caulker Marine Reserves. Coral Gardens is a system of shallow water (<7 m) patch reefs dominated by ~75,000 m$^2$ of *A. cervicornis* and *A. palmata*, and it contains one of the largest recorded acroporid populations in the Caribbean (Fig 1; [27]). All three Caribbean *Acropora* species (*A. cervicornis*, *A. palmata*, and *A. prolifera*) are present at Coral Gardens in substantive thickets that are interspersed between mixed massive and finger corals (e.g., *Millepora*, *Porites*, *Orbicella*, and *Agaricia* spp.) and sandy areas. The branching nature of acroporids provides habitat to a number of herbivore species, notably *Echinometra viridis* urchins and several species of damselfish (e.g., *Stegastes adustus*, *Stegastes planifrons*, and *Microspathodon chrysurus*). The mostly monospecific thickets of branching acroporids vary in size and can reach 35 m in diameter. The study site has periodically experienced the passage of tropical storms and hurricanes, but it is not located in the outflow path of any major rivers. Rainfall and SST variability at Coral Gardens are presumed to be roughly similar to other sites in Belize where acroporids are less abundant.

## Materials and methods

### Site selection

Coral Gardens was first surveyed in 2011 when reconnaissance studies were conducted across the wider perimeter of the site in search of monitoring locations. The first transect site (T1) was established and marked with rebar stakes placed in hardground or non-living coral framework (Fig 1). To best represent potential variability across the reef and collect data from a wider area, four additional transects (T2, T3, T4, and T5) were established in 2012. All five

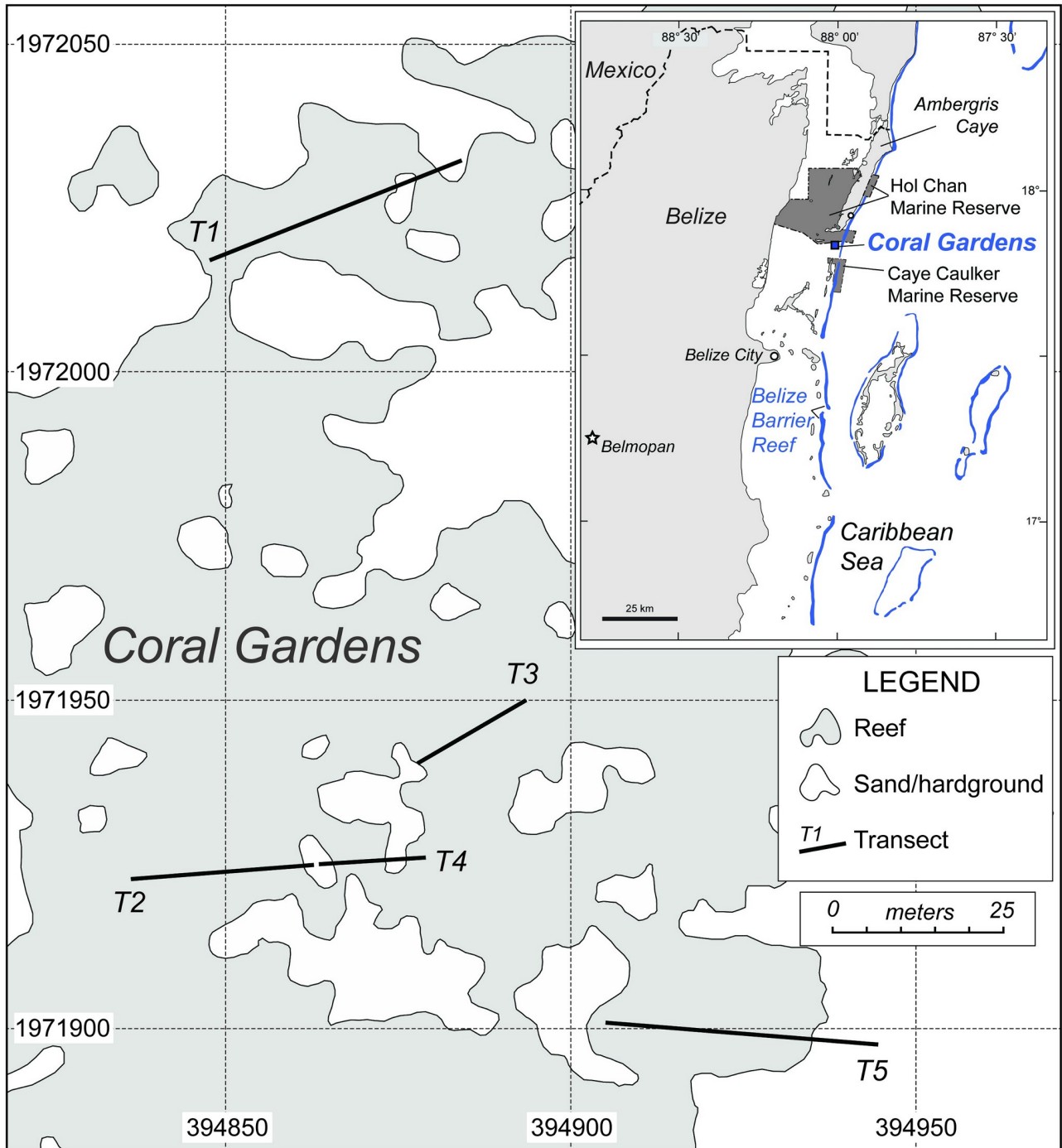

**Fig 1. Index map for Coral Gardens including locations of transects used in this study.** Coordinates are UTM northings and eastings in Zone 16. Modified from [16].

transect locations were documented with high-resolution GPS measurements taken using a Geoexplorer 6000 series Trimble GPS unit. Transects vary in length from 12–36 m. To explore patterns in live coral cover over time, we treated each transect as an independent replicate sample from a distinct coral patch. Live coral cover in each transect was calculated as the

average of quadrat samples within the transect (see below). Mean annual coral cover at Coral Gardens was calculated as the average of the five transects, with each transect weighted equally.

## Field documentation

*Acropora cervicornis* was surveyed at all five transect locations using the same protocol from 2012–2019 during each summer (excluding 2015) for seven years. Transects 1 and 5 were also surveyed in October 2016. To survey coral abundance quantitatively, a measuring tape was placed between transect markers at the beginning and end of each transect. Using a 1 m² scale quadrat constructed from PVC pipes, multiple photographs were taken from overhead (map view) at 1 m intervals by divers using at least two cameras to optimize image quality options for post-processing and in case of camera malfunction (Fig 2). This process was replicated at each meter along each transect. A variety of underwater cameras were used over the years, including Panasonic Lumix, Olympus Tough, FujiFilm XP, and Nikon Coolpix. In 2013, the perimeters of *A. cervicornis* patches were mapped using the handheld Geoexplorer 6000 Trimble GPS unit by having a small boat closely follow a swimmer (within ~3 m) who followed the contour of all areas of the reef (Fig 1). All necessary permits (CITES and Belize Marine Fisheries Unit) were obtained for the collection of samples for dating and permission for field photography, and we complied with all relevant regulations. Permits include Permit #000033 (and renewals) from Belize Fisheries Department Ministry for Forestry, Fisheries and Sustainable Development, CITES International Trade in Endangered Species of Wild Fauna and Flora Permit No. 5673 and Port Entry number 2016075 after compliance review (Dec Control Num 2016876077).

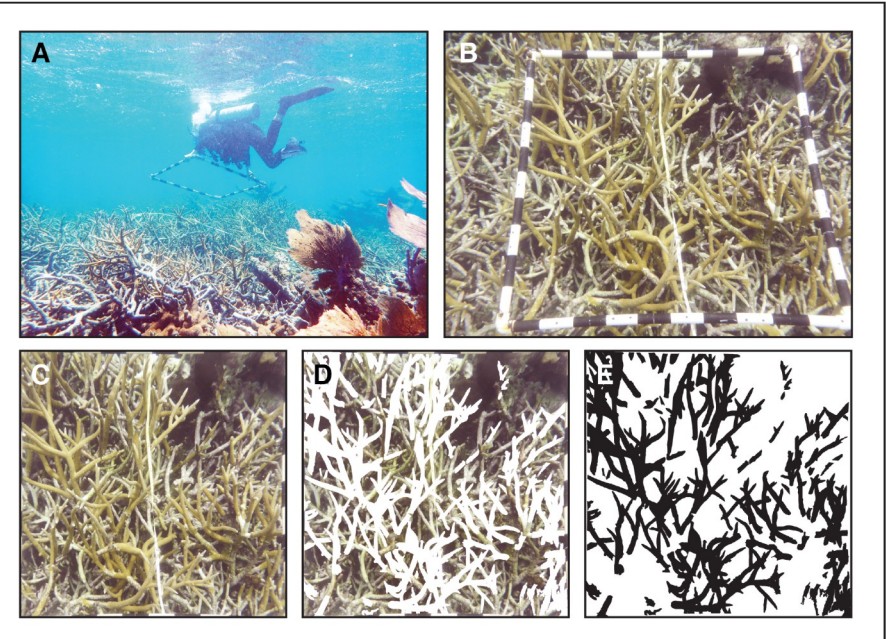

**Fig 2. Live coral assessment methods.** (A) Diver placing a 1 m² quadrat above reef acroporids; (B) Raw image taken from above the quadrat; (C) Image rectified (cropped and stretched) using a MATLAB script; (D) Same image with live coral manually traced using Adobe Illustrator; (E) Same area (live coral now in black) that was used to quantify the area of live coral tissue per 1 m² using a MATLAB script.

## Image processing

Multiple photographic images were obtained for each individual 1 m² quadrat and assessed for angle, focus, lighting, and spatial completeness (showing all four corners of the quadrat scale). Images that best enabled discrimination between coral species, live coral, encrusting algae, macroalgae, and other hard rubble substrates was selected for processing. Individual quadrat images were cropped and rectified using MATLAB software, with scripts developed by Stough et al. [46] (Fig 2). The calibrated photographs were manually segmented (traced) by individual evaluators using Adobe Illustrator (Fig 2). Evaluators outlined all live *A. cervicornis* tissue by using the Pen Tool to click along the contours of living branches and enclose all visible live coral within polygons that were then filled in solid black. A white rectangle in a separate, posterior layer was used to create a completely black and white image which was exported as a Portable Network Graphic (PNG) image file (Fig 2). A MATLAB script was used to calculate percent live coral (i.e., pixel percentage of black in the image). Every completed image was assessed and edited by a second evaluator for completeness and accuracy to ensure consistency between evaluators. Inter-evaluator variability was assessed during training until a standard deviation of less than 0.01% per image was achieved. Because photographs only represent a two-dimensional map view of three-dimensional branching thickets, live coral quantified by this method is a conservative measure of actual live *A. cervicornis* tissue biomass but well represents 2D coverage on the reef [47].

## Temperature

Temperature data were collected at T1 and T5 in Coral Gardens at 1 to 60- minute intervals from 5/9/12 to 7/2/19 using HOBO Pendant Temperature and Temperature/Light Data Loggers (UA-001 and UA-002) manufactured by the Onset Computer Corporation. Loggers (sensors) were deployed 2–3 m below mean sea level and attached with zip ties to non-living reef substrate adjacent to *A. cervicornis* thickets. Multiple sensors were deployed each year and compared for cross calibration and/or drift. A single time series was constructed from validated sensors. SST data recorded from 0600–0700 hours were averaged to daily, monthly, and annual means for data analysis.

The *in situ* sensor data was augmented with 0.5-deg gridded SST data (*CoralTemp v3.1*) from the Coral Reef Watch 5-km product suite [48], and heat stress metrics were calculated following the methods of Liu et al. [49] as outlined below. First, we compared the mean temperatures for each month during the sensor/gridded overlap period (May 2012—June 2019) to ascertain the equivalence of the *in situ* and satellite datasets. This showed that sensor ($SST_{sensor}$) and gridded ($SST_{gridded}$) SST's were highly correlated ($R^2 = 0.998$), and that the gridded data show a small negative bias relative to the sensor data ($SST_{gridded} = 0.9557$ x $SST_{sensor} + 1.0842$). Degree Heating Weeks (DHW), a commonly used metric for heat stress exposure in corals, is defined as the twelve-week accumulation of daily SST's that are 1˚C warmer than maximum monthly mean (MMM) temperature [34]. We calculated MMM for Coral Gardens using the gridded data for 1985–1990 + 1993, which yielded a MMM temperature of 28.85 ˚C during the baseline period. The baseline MMM was then bias corrected to 29.04 ˚C based on the comparison of $SST_{sensor}$ and $SST_{gridded}$ during the period of sensor/gridded SST overlap. Daily HotSpot values (HS) were calculated using the daily sensor SST and the bias-corrected MMM (HS = SST–MMM when SST > MMM). Degree heating-weeks were calculated by summing the daily HS values that exceed the bleaching threshold (MMM + 1˚C) over a 12-week period.

## pH

In 2019, sixty-four temperature and pH measurements were collected with two Oakton PTTestr 35 instruments at 1 m depth using a LaMotte water sampler. A three-point calibration was performed daily, and samples were collected over a five-day period (June 29-July 3, 2019) at four locations off the southern end of Ambergris Caye. Samples were collected at least 50 m from shore to avoid the localized effects of decomposing *Sargassum* sp. near shore that in recent years has become an annual spring and summer event. The mean pH of 8.1 (n = 64) was obtained, with a range of 7.7–8.3. Only eight measurements fell below a pH of 8.0 (S1 Table).

## Sediment, substrate and macroalgal abundance

A point count method was used to estimate the composition of substrate along the center of each transect at 0.5 m intervals from all quadrat photographs in 2014 and at the same sample interval in the field in 2019. Live coral, bare rock, sediment, and macroalgae were noted by two evaluators and averaged, with the number of points per transect serving as a low-resolution proxy for percent substrate. A thin steel measuring probe was used to measure canopy height, sediment thickness, and depth to hardground along transects 1 and 5 in 2014.

## Herbivory

Urchin counts were performed in the field for every transect quadrat in both 2014 and 2019. All urchins were counted twice per $m^2$ by two evaluators and averaged. Over 99% of the urchins counted were identified as *Echinometra* spp. Visual surveys of damselfish (*Stegastes adustus*, *Stegastes planifrons*, and *Microspathodon chrysurus*) were conducted in 2014 and 2019. Fish were observed for every $m^2$ of the transects for ~1 minute by two evaluators hovering at least 1 m above the substrate. Fish were recorded if they occupied the area for over 3 seconds and individual fish species were not differentiated.

# Results

## Live coral abundance

In total, over 1,000 individual quadrat images were collected between 2012–2019 at the five transect sites. Over the course of the study, live coral coverage declined at all transect locations (Table 1, Fig 3A, S2 Table). Between 2012 and 2019, percent live coral cover (calculated as an average of the five equally weighted transects) declined by 16.2% per $m^2$. Significance of

**Table 1. Number of 1 $m^2$ quadrats and percent mean live coral abundance per year from 2012–2019 at Coral Gardens.** Transect locations are indicated with T1-T5, and n is the number of 1 $m^2$ photographed quadrats. The mean is the percentage mean of five equally weighted transects.

| Year | n | T1 | T2 | T3 | T4 | T5 | Mean |
|---|---|---|---|---|---|---|---|
| 2012 | 133 | 19.61 | 29.69 | 24.21 | 23.73 | 58.05 | 31.06 |
| 2013 | 129 | 19.31 | 25.98 | 23.05 | 32.68 | 56.16 | 31.43 |
| 2014 | 130 | 14.54 | 28.28 | 21.05 | 19.39 | 51.67 | 26.99 |
| 2015 | 0 | | | | | | |
| 2016 | 131 | 13.70 | 28.78 | 13.39 | 22.52 | 35.55 | 22.79 |
| 2016 Oct | 67 | 6.63 | | | | 18.65 | 12.64 |
| 2017 | 133 | 8.70 | 15.81 | 6.97 | 22.27 | 18.09 | 14.37 |
| 2018 | 140 | 9.85 | 16.48 | 4.63 | 16.45 | 19.52 | 13.39 |
| 2019 | 141 | 12.31 | 14.92 | 5.33 | 16.07 | 25.48 | 14.82 |

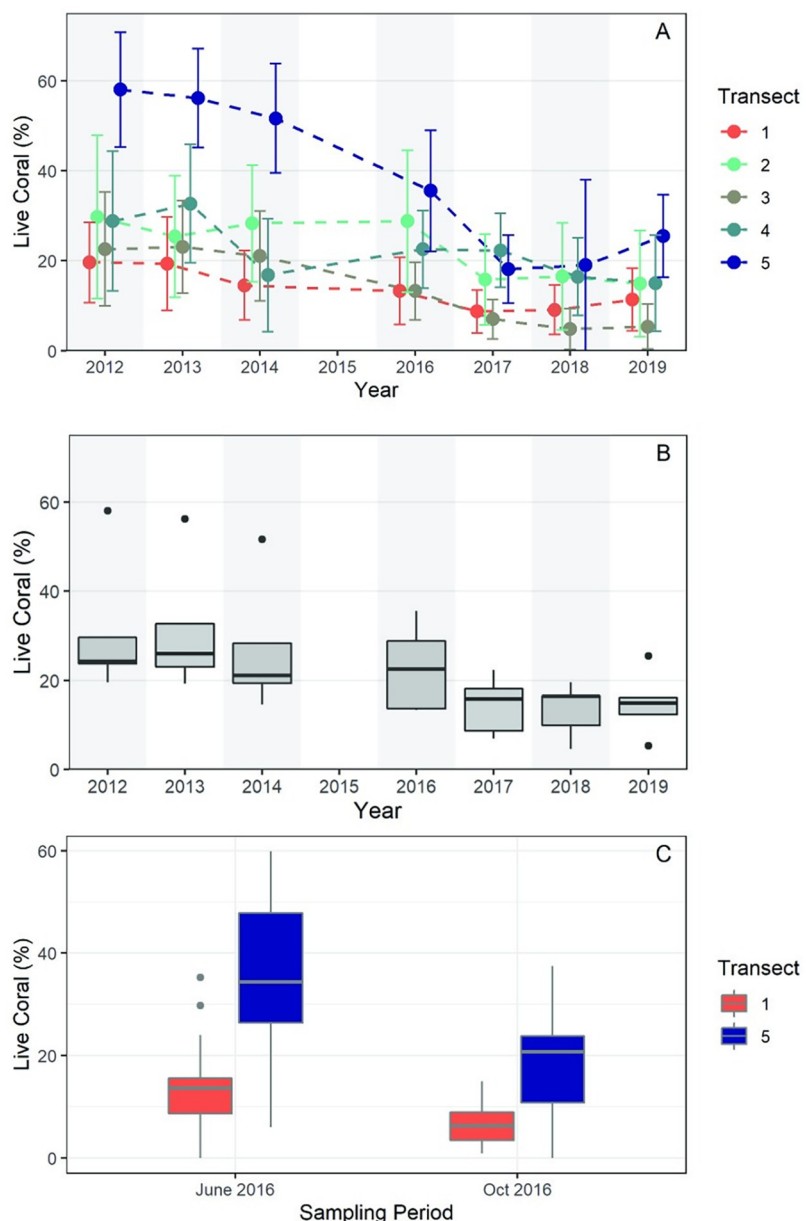

**Fig 3. Summary of mean live coral per 1 m² at each Coral Gardens transect site from 2012–2019.** (A) Mean percent live coral per year per transect with 1-sigma errors; (B) Boxplot of live coral percentage as an average of the five equally weighted transects shown by year. Shaded rectangles indicate the interquartile range, whiskers indicate minimum and maximum values, the thick horizontal lines are at median values, and filled circles are statistical outliers; (C) Boxplot of live coral at Transects 1 and 5 in June and October of 2016. Symbols as in 3B. A and B do not include October 2016 data.

the temporal decline was confirmed by repeated-measures ANOVA (P<0.01). The site with the most abundant live *A. cervicornis* (T5) suffered a loss of 32.6% and serves as an outlier among the replicate transects (Fig 3A), with the greatest loss between 2016 and 2017. Pairwise comparisons of sequential annual means indicate that only the difference in live coral cover between 2016 (mean = 22.8, SD = 9.6) and 2017 (mean = 14.4, SD = 6.4) is significant (pairwise t-test with sequential Bonferroni adjustment; $P = 0.05$). Only two sites (T1 and T5) were

**Table 2. Summary of annual (calendar year) temperature data at Coral Gardens.**

| Year | Mean Annual Temp ˚C | Start of Sustained SST>MMM | Days SST>MMM | Weeks DHW>0 | Peak DHW (˚C week) |
|------|---------------------|----------------------------|--------------|-------------|--------------------|
| 2013 | 28.23 | 27-Jul | 93 | 24.4 | 2.3 |
| 2014 | 28.20 | 16-Jul | 88 | 16.0 | 2.0 |
| 2015 | 28.42 | 27-Jul | 108 | 21.7 | 6.8 |
| 2016 | 28.67 | 18-May | 168 | 32.1 | 5.4 |
| 2017 | 28.47 | 16-Jun | 136 | 29.6 | 7.9 |
| 2018 | 28.24 | 17-Jul | 105 | 23.6 | 1.3 |

Although temperatures were generally warmer throughout 2015–2017, SST's increased earlier and remained elevated longer during 2016.

surveyed in both summer and fall of 2016. Both sites showed a significant decline in live coral between June and October of that year (Fig 3C and Table 1), with a mean loss of live coral tissue of 7.1% at T1 and 16.9% at T5 and an overall decline of 11.9% living coral for the two sites. Live coral tissue declined at all but one comparable quadrat along T1 and T5 over the four months.

## Temperature

The hottest year on modern record prior to the end of this study occurred in 2016, an El Niño year, with global temperatures 1.0˚C above the 20th century average for land and ocean temperatures combined, and 0.79˚C above average ocean temperature (SST; [50]). That year was also the hottest year on average at Coral Gardens (Table 2, Figs 4 and 5, S3 Table). Sea Surface Temperatures at Coral Gardens exceeded the baseline (1985–1990+1993) MMM in each of the years of the study (Fig 4), although with considerable variation. During 2012–2014, daily SST's only exceeded the MMM+1 ˚C threshold by small amounts and for relatively brief periods of time, resulting in maximum DHW of <2.3 ˚C-weeks. By comparison, 2015, 2016, and 2017 were significantly warmer, with peak DHW's reaching 6.8, 5.4, and 7.9 ˚C-weeks respectively, and exceeding the "bleaching risk" ($\geq$ 4 ˚C-weeks) threshold in each of the three years (Fig 4). The year 2016 did not accumulate the greatest DHW observed during the study period, but is remarkable in that SST's became elevated above MMM far earlier (mid-May, as compared with mid-July) and continued for the longest duration (>5.5 months) of the study period.

## Sediment analysis, substrate, and bathymetry

The seafloor sediment is composed entirely of carbonate and organic material ranging from *in situ* micrite to coarse sand-sized skeletal and calcareous algae grains. Sediment surrounding coral thickets is coarse, with fines (<64 μm) not exceeding 2% of the cumulative sediment sample weight at any location. Sediment thickness above hardground averaged roughly only 2–3 cm in depth at 10 locations proximal to transect locations and rarely exceeded 8 cm thickness in small pockets around the perimeter of the *A. cervicornis* patches (S4 Table). Sediment thickness beneath coral canopy was generally on the order of a cm or less. Coral canopy height varied from ~3.9–1.7 m below mean sea level at transects 1 and 5 in 2014, and depth to hardground varied from ~4.3–2.8 m below sea level, resulting in an approximate average coral canopy height of 0.7 m above substrate at T1 and 1.1 m above substrate at T5 (S5 Table).

## Macroalgae and herbivory

*Echinometra* spp. urchins are abundant at Coral Gardens. The mean number of urchins per m$^2$ was 20.4 (SD = 14) in 2014 and 24.1 (SD = 13.5) in 2019 (S6 Table). The average damselfish

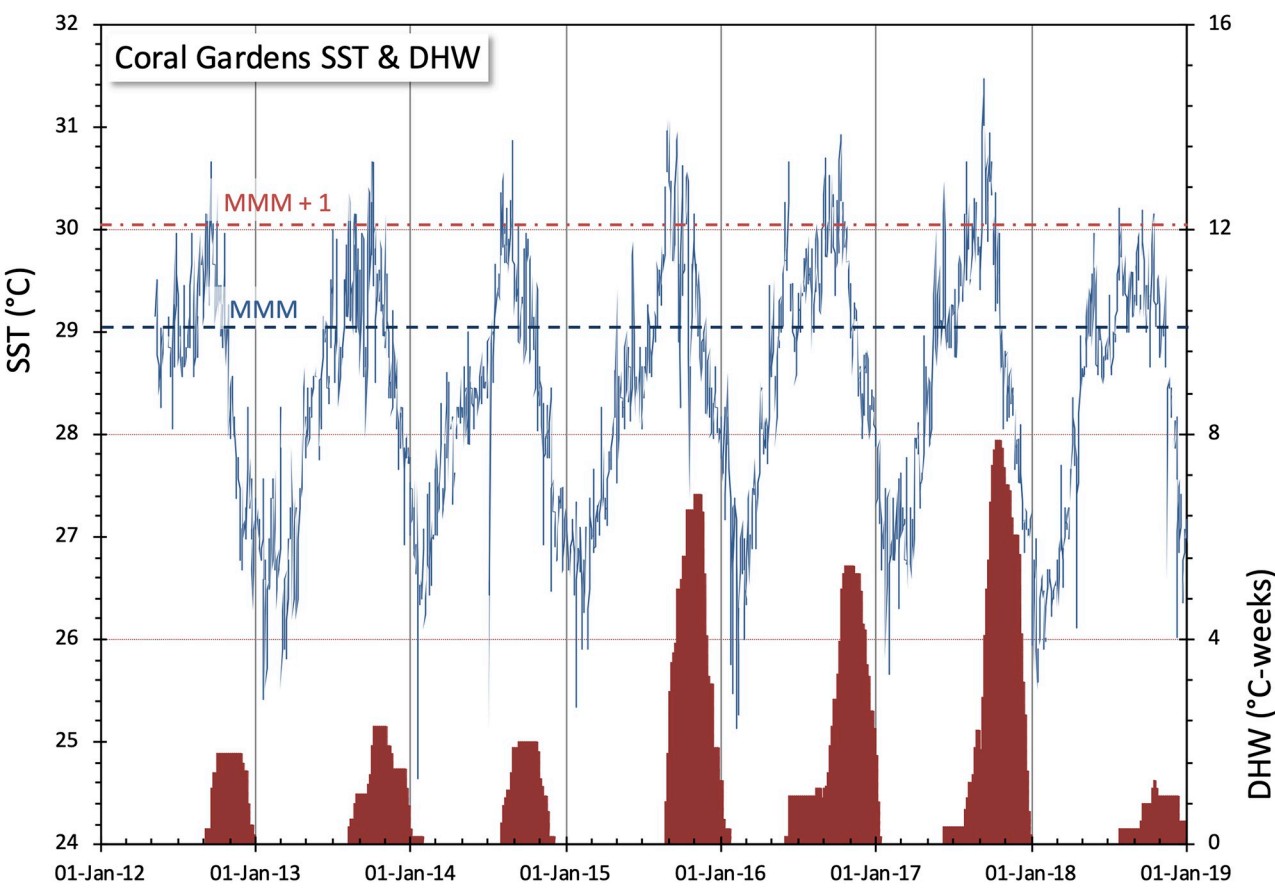

**Fig 4. Mean daily SST and accumulated degree heating week at Coral Gardens from 2012–2019.** Sea surface temperatures (thin blue line) are from in-situ sensors. Degree Heating Weeks (shown in red) accumulate when SST's exceed the MMM + 1°C threshold. See text for details.

abundance was 2.0 (SD = 1.3) per m$^2$ in 2014 and 2.3 (SD = 1.2) in 2019 (S7 Table). Statistical comparisons indicate these are not significant differences. A total of 284 points were evaluated for substrate composition along the transects in 2014 and 2019 (S8 Table). There was no significant change in macroalgae abundance between 2014 (mean = 5.4%, SD = 2%) and 2019

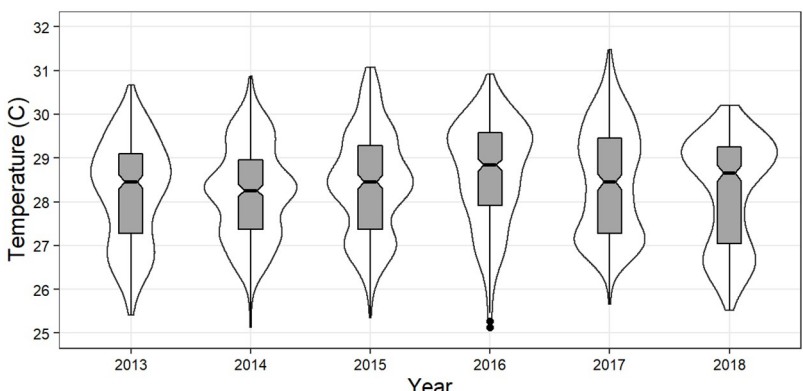

**Fig 5. Summary of temperature data in form of a violin plot of daily average temperature by year with inset box plots.** Outer lines depict distribution of observations within the sample year. Box plots depict the standard quartiles and data range.

(mean = 4.5%, SD = 3.7%) ($P$ = 0.66) although point count data indicate a significant decline in *A. cervicornis* between these years (pairwise t-test; $P$ = 0.03).

## Discussion

Live *Acropora cervicornis* cover declined significantly at Coral Gardens between 2012–2019. While live coral abundance decreased overall at every transect, healthy *A. cervicornis* colonies remained at each of the five transects. The greatest loss in total live coral tissue in the study period occurred between June 2016 and June 2017. This loss follows a rise in DHW in 2015, and coincides with a period when temperatures warmed earlier, stayed higher, and lasted longer than any of the other years in this study (Table 2, Figs 4 and 5). Coral loss also coincides with the passage of Hurricane Earl in August 2016, although significant physical damage from the storm was not observed. Below we discuss temperature, storms, and additional potential factors contributing to both the decline and persistence of *A. cervicornis* at Coral Gardens and, by process of elimination, attempt to discern the most likely factors for changing coral abundance in this unique refugium.

### Candidates for *Acropora cervicornis* decline at Coral Gardens

**Herbivory, macroalgal abundance, and predation.** The detrimental effects of macroalgal occupation on potential recruitment and growth of corals has been well recognized [37, 38]. Urchins and damselfish are important herbivores that feed on the algae that covers reef framework, and serve as a critical control on macroalgal abundance [3, 51]. Here, we found no evidence of a significant change in macroalgal abundance, urchin density, or damselfish density at Coral Gardens during the study period. Although no nutrient data was collected, the reef site is not adjacent to large tracts of fertilized farmland, point-source pollution, or riverine influx. The 2017 IUCN Report [26] lists water pollution as a low threat to the Belize Barrier Reef, and the low percentage of macroalgae substrate and lack of an increase in algae over the study period indirectly support the conclusion that eutrophication was unlikely to be a leading factor in coral decline.

Excess predation and bioerosion are also potential problems for acroporid reefs. Urchins and damselfish may contribute to bioerosion and grazing of live coral tissue, but they also provide superior habitat for coral settlement by providing clear space for coral recruits, so the benefit to corals is usually thought to outweigh cost in terms of space provision [52, 53]. Even if bioerosion was prevalent at Coral Gardens, the web-like canopy for *A. cervicornis* 'props up' coral branches that have become detached at the bottom, and no canopy collapse was observed during the study. Predators known to cause tissue damage to acroporid corals include fireworms (*Hermodice carunculata*) and prosobranch gastropods (e.g., *Coralliophila abbreviata*; [54]), but fireworms and prosobranch gastropod sightings have been rare to absent at the transect locations (Greer, personal observation) during the study. We conclude that there is no evidence to support predation as a driving factor for coral decline at Coral Gardens.

**Sea level rise, ocean acidification, and White Band Disease.** The decline of coral cover at Coral Gardens cannot be attributed to current rates of sea level rise. *A. cervicornis* is one of the fastest growing Caribbean corals with linear extension rates of 3–11 cm per year that well outpace current rates of sea level rise [4, 5]. Although ocean acidification is increasingly suggested to provide future challenges for reefs, as well as for certain reefs at present [55], there is not clear evidence to suggest that acidification already poses a threat to live coral tissue or is currently a threat to all reefs [56]. We do not have long-term pH data for Coral Gardens, but average measurements of 8.1 in June 2019 mirror current normal marine values.

White Band Disease (WBD) has been devastating to acroporids across the Caribbean [45]. While we personally cannot accurately distinguish WBD from bleaching of acroporids without histopathology, visual observation of white *A. cervicornis* (i.e., clear tissue revealing white skeleton beneath) at Coral Gardens was minimal in all years of the study period, with few potential WBD cases observed on, or in close proximity to, the actual transects. This does not mean that bleaching or WBD were entirely absent and we recognize the limited 'snapshot' scope of study. We only note that evidence of WBD or bleaching was only very rarely observed during data collection periods.

**Storms.** No two tropical storm events are exactly alike and consequently the impacts to reefs may vary in nature and scope. The direction of travel, distance from storm epicenter, intensity, duration, and storm surge all govern impact effects. This means that the 'category' of the storm is not the only measure of influence to a reef site. While not perfectly correlative, major storm and hurricane events can damage coral reefs in several ways. Excessive inland rain can cause freshwater flooding (i.e., salinity stress), the influx of terrestrial organic matter and clouding of water (i.e., low light), and runoff of contaminants like fertilizers from farmland to reef (i.e., nutrient influx); all of which can adversely affect adjacent coral communities [40]. Coral Gardens is not located near significant riverine or land-based runoff sources and is therefore unlikely to be impacted by runoff. The physical impacts of wave damage can result in reef flattening, especially of the shallowest coral stands that reach closest to the water's surface. Even when canopy collapse does not occur during a storm, loose branches may experience some abrasion if mobilized [57]. No canopy collapse or major structural damage was observed at Coral Gardens during the study period.

To explore whether storms have resulted in recognizable mortality events prior to the study period at Coral Gardens, we compared $^{230}$Th dates from a previous analysis of dead corals excavated from Coral Gardens ([16]; Fig 6, S9 Table) with known storm events that passed in close proximity from 1980–2019 [58, 59]. Ages were obtained using methods from Clark et al. [60, 61]. The $^{230}$Th dates from dead coral samples revealed peaks in ages at around ~1994 and ~2009. No tropical storms or hurricanes passed near Coral Gardens from 1981–1998, or in 2009. Tropical storms and hurricanes with potential for damage did pass in close proximity to the site in 1980, 1998, 2000, 2001, 2007, 2008, 2010, 2012, 2016, and 2017. However, there appears to be no clear correlation between known storms events and coral mortality dates.

Strong storms passed near Coral Gardens with no coincident mortality dates, yet other dates suggest mortality coincident with milder storms or no storms at all. No tropical storms or hurricanes impacted the field area in Belize between 1981 and 1998 which bracketed 14 of the 26 $^{230}$Th ages (Fig 6). Hurricane Mitch, one of the deadliest Atlantic hurricanes on record, passed south of Belize as a Category 5 storm in 1998. Extensive damage to reefs was reported at Glovers Atoll to the south, with up to 90% mortality of *Acropora palmata* [62] at the site. Personal storm accounts reported significant hurricane damage to infrastructure and the coastline of Ambergris Caye with high waves at the reef crest. Furthermore, Hurricane Mitch was coincident with a significant bleaching event [62]. Yet no $^{230}$Th ages coincide with this year. In 2000, Hurricane Keith arrived just offshore of Ambergris Caye as a Category 4 hurricane with peak winds of 225 km/h on October 1, causing considerable damage on land [63]. No coral $^{230}$Th ages coincide with this event and only one does so in the next year. In 2007, Hurricane Dean landed in Mexico about 65 km north of Belize as a Category 5 storm, and major damage was reported in Corozal [64], but the San Pedro Sun reported that "Early estimates and initial hotel reports indicate very minimal damage to areas such as San Pedro and Caye Caulker" (https://ambergriscaye.com/sanpedrosun/old/07-331.html). Tropical Storm Arthur arrived in northern Belize from the Pacific in late May of 2008, reportedly causing heavy surf near Ambergris Caye [65], but no major damage. No coral mortality dates match

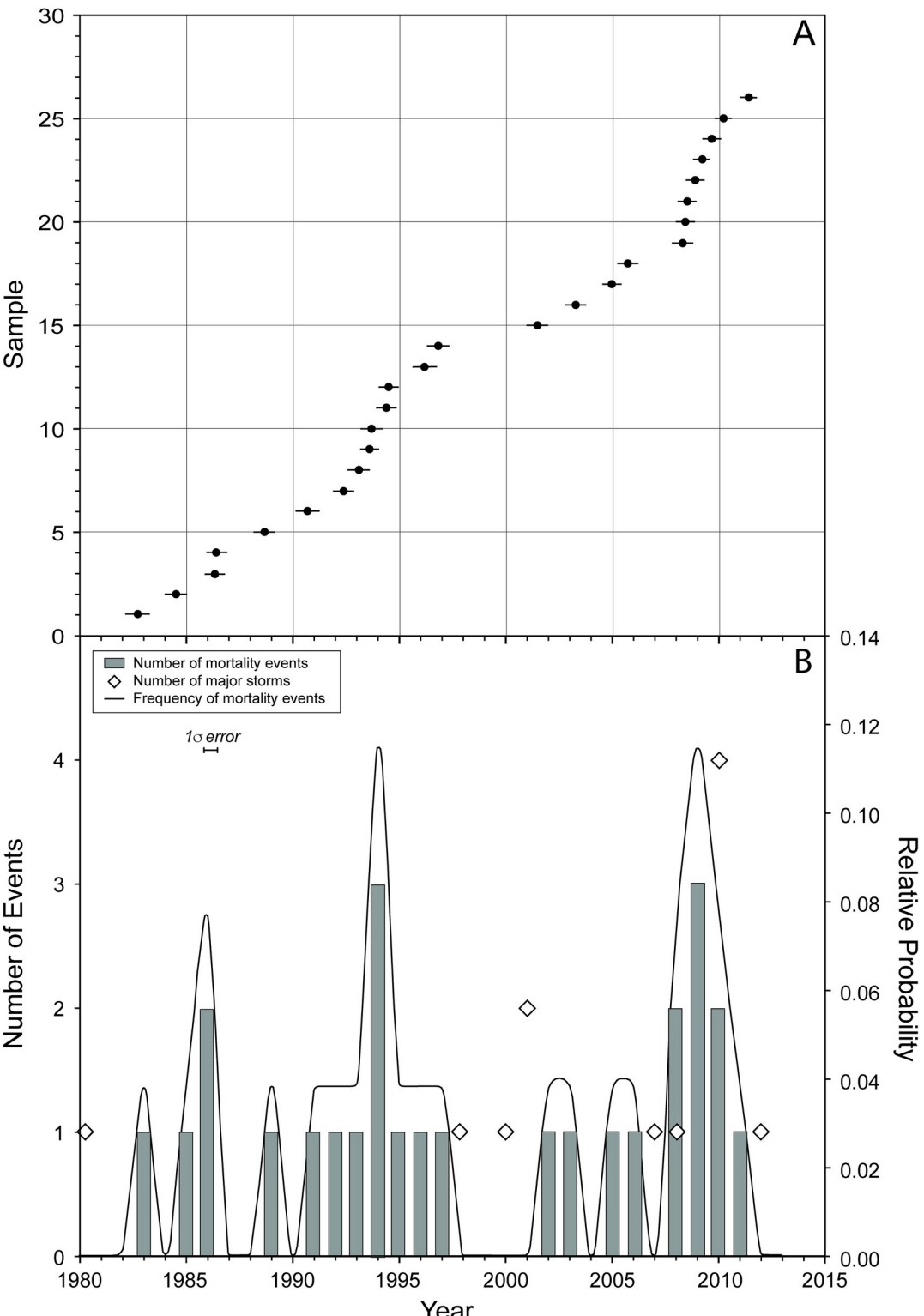

**Fig 6. *Acropora cervicornis* <sup>230</sup>Th mortality events and known storms.** (A) *Acropora cervicornis* <sup>230</sup>Th ages (±1 σ error bars) determined from 26 coral fragments collected from the substrate of transect T5 and excavated proximal to the transect. In this plot, significant mortality events would be indicated by several samples with the same or similar ages (e.g., forming a horizontal array); (B) Histogram and relative probability curve of *Acropora cervicornis* <sup>230</sup>Th mortality events. The relative probability curve (black) shows peaks in coral mortality centered at 1986, 1994, and 2009. Diamonds indicate number of known tropical storms and hurricanes in proximity of Coral Gardens in a given year.

2007, but 3 of the T5 coral samples died in 2008, and two the year after when no storms passed Coral Gardens.

In 2016, Tropical Storm Earl strengthened to a Category 1 hurricane with 147 km/h winds before making landfall about 10 km south of Belize City on August 4 [66]. This storm caused damage to docks and coastal infrastructure on Ambergris Caye with heavy rainfall and maximum storm surge estimated to be between 1.2–1.8 m [66, 67]. Yet the coral canopy remained intact at Coral Gardens and significant structural damage (reef flattening, denuded patches along the transects) was not apparent. Live coral declined across all areas of the reef with the decrease in abundance distributed across all areas of the surveyed sites (cover declined in all but one individual m$^2$ in T1 and T5 from June to November) in 2016. The only other storm to pass near Coral Gardens was Tropical Storm Franklin, which passed just north of Belize on August 8, 2017 [68]. Again, no structural damage was noted at the site. Although Hurricane Earl cannot be ruled out as a contributing force for coral mortality between the June and October surveys in 2016, precedent suggests that *A. cervicornis* at Coral Gardens persisted despite close proximity to several tropical storms and hurricanes in the recent past.

**Sedimentation.**   Excess sedimentation or mobilization of sediment can pose a significant threat to coral reefs [69]. Little loose sediment is present at Coral Gardens and most of the sediment is not fine enough to remain in suspension for long under normal conditions. The sediment surface sits 3–4 m below the surface at Coral Gardens, and the coral canopy rises between 0.7–1.1 m above substrate on average, so it is unlikely that sedimentation stress is a driving threat to the corals under ordinary conditions. However, sediment can be resuspended at great depths during storm wave circumstances [70], potentially leading to abrasion of coral tissue and an energetic cost of sediment removal for the coral polyps.

**Additional threats to reefs.**   Recreational activity can certainly be detrimental to the health of a reef and has been cited as a contributing factor to the threatened status of *Acropora* spp. reefs [3]. Anchor and chain damage, vessel groundings, careless snorkelers or divers, lost fishing gear, and interference in herbivore population densities can create physical as well as ecosystem damage unfavorable to coral growth, especially for branching *Acropora* spp. in shallow habitats [3, 71]. Tourist visitation metrics have not been assessed at Coral Gardens, but there were no reported groundings in the vicinity during the study period, and we did not observe any obvious structural damage at the transect locations. Although Coral Gardens is visited by commercial and recreational fishers and snorkelers, the area is situated within a few kilometers of more established recreation sites, including the Hol Chan Marine Reserve, Shark and Ray Alley, and the Caye Caulker Marine Reserve, all of which draw heavier tourist attention than Coral Gardens.

**Temperature.**   It does not appear that the factors discussed above are likely to be significant drivers of coral decline at Coral Gardens, with the possible exception being sediment-derived abrasion of live coral tissue during Hurricane Earl. Using a heuristic process of elimination, this leads to temperature as a leading candidate contributing to mortality of *A. cervicornis* at Coral Gardens during the study period. The coincidence of both anomalous temperatures and the passage of Hurricane Earl in 2016 may be especially significant. Both the intensity and the duration of heat anomalies can contribute to coral stress [34]. Data from Coral Gardens show that 2016 warmed sooner and SST's remained higher for longer than other years in the study period, resulting in 168 days above MMM and an accumulation of 32.1 DHW (Figs 4 and 5).

While we did not observe bleaching during our field visits to Coral Gardens, we cannot rule out bleaching as a contributor to coral demise, as we have only annual data of coral cover for each transect site. It should also be noted that corals and their algal symbionts are physiologically subject to heat stress prior to the onset of outward physical expression, as evidenced by

certain biomarkers for heat stress prior to bleaching [72–74]. Genetic markers linked to coral metabolism, heat shock protein response, immune response, and the transport of $Ca^{2+}$ across cell membranes (key to calcification) have been documented [72, 74]. During thermal stress, photosynthetic dysfunction can damage nucleic acids, oxidize cellular membranes, and denature proteins within the cell [75, 76]. Paradis et al. [77] showed that important physiological changes in the ratio of photosynthesis to respiration (P:R) occur in *A. cervicornis* corals with increasing temperature above the thermal optimum. They measured a significant decline in net photosynthesis at 28˚C and 30˚C temperatures (in their mesocosm study, optimal P:R was at 25˚C), and respiration increased linearly with temperature. They noted that these physiological changes at high temperature were visually undetectable prior to bleaching but resulted in a negative energy balance for the corals. They also noted that even when heat-stressed corals retain algal symbionts, they may have already entered a state of malnutrition, which could lead to increased vulnerability to additional stressors.

With no visual indication, heat stress might be accumulating when temperatures are greater than optimal, even if below accepted "threshold" temperatures [77]; however, it is not easy to determine optimal temperature in a field setting because corals in different locations may be acclimatized to different temperatures. We note that live coral at Transect 5 declined between 2014 and 2016, preceding the more widespread decline observed 2016–2017 (Fig 3A). While we cannot determine exact timing of coral loss between 2014–2016 due to a lost field season, it is possible that corals from Transect 5 were more susceptible to the onset of a marked increase in days above MMM and DHW accumulation in 2015 than at other field locations (Fig 4). It is certainly possible that partial bleaching did occur in that time. While Belize corals have been known to recover from partial bleaching in the past [62], perhaps 2015 marked the onset of thermal stress at Transect 5, and 2016 temperatures further impacted the Coral Gardens sites.

Paradis et al. [77] also documented a dramatic compounding impact on the ratio of photosynthesis to respiration (P:R) in *A. cervicornis* corals subject to abrasion. They inflicted corals with wounds from ~1.5–5.0 cm$^2$ in size and compared the metabolic responses of abraded versus non-abraded corals. Their results showed that photosynthesis decreased dramatically, and significantly more in abraded corals at higher temperatures (28˚C and 30˚C), with up to 90% less oxygen production, even in non-damaged coral tissue in the wounded specimens. While the authors abraded corals as a proxy for diver damage, we suggest it is possible that storms can also cause abrasion damage at Coral Gardens and elsewhere. The energetic cost of sediment removal from coral tissue may compound this impact. Tropical storm and hurricane sediment mobilization may also cause abrasion even when the structural foundation is maintained. In addition to sediment suspension, we have observed that even when architectural integrity remains sound, loose branches may move and touch each other during high wave activity. Under optimal temperatures corals may be able to heal, but at higher temperatures they might not. We suggest that although storms may not have resulted in recognizable dated mortality events in the past at Coral Gardens, temperature conditions in 2016, and possibly even 2015 at Transect 5, could have resulted in an increased vulnerability of corals to the passage of Hurricane Earl.

## Resilience, persistence, and the refugium concept

Despite the decrease in overall live coral cover from 2012 to 2019, *A. cervicornis* cover stabilized at Coral Gardens after 2016, and statistical comparisons indicate no significant change in live coral abundance after 2017. The lack of coral decline after 2016–2017 coincided with lower mean annual SST values in 2018, and a marked decrease in days above MMM and accumulated DHW (Fig 4). In 2019 there were still areas of Coral Gardens with abundant *A.*

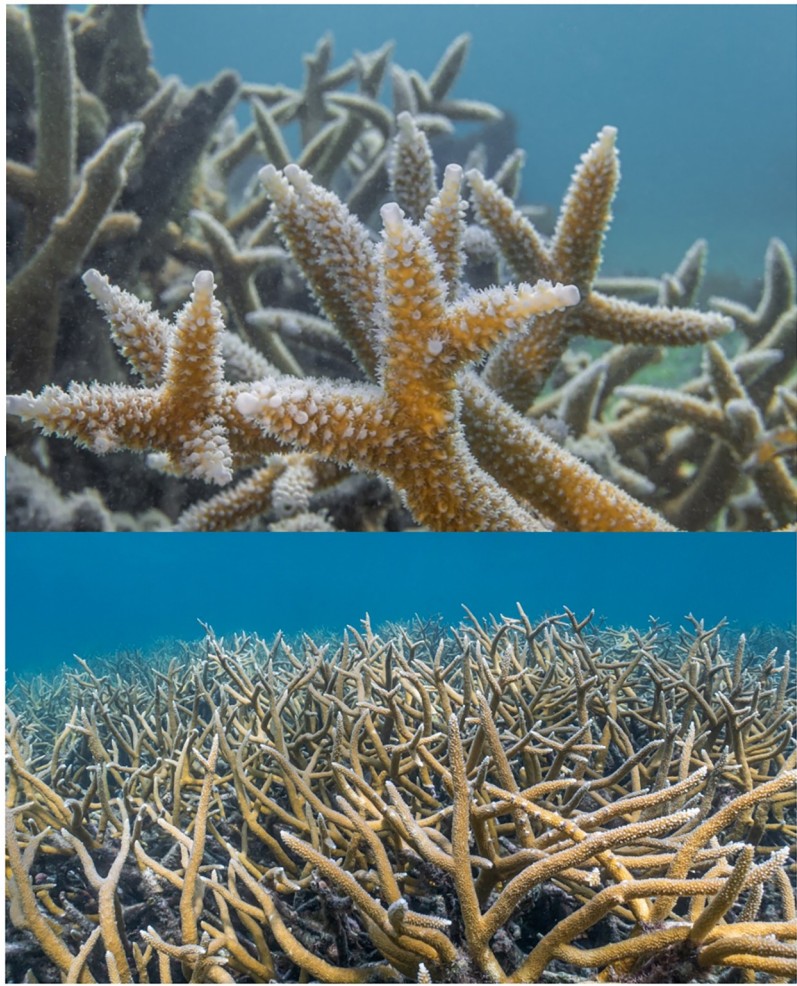

**Fig 7. Healthy *Acropora cervicornis* at Transect 5 in June 2019.**

*cervicornis* coral cover that exceeded reported abundances in most areas of the Caribbean that we know of (Fig 7). The very lack of the coral stressors discussed here, excluding temperature, and perhaps in combination with a passing storm, may in part explain the apparent resistance of *A. cervicornis* to collapse (even if compromised in 2016) at Coral Gardens.

Connectivity to adjacent vibrant, healthy, or protected reef areas can be an important factor in reef health [42]. Coral Gardens is situated between two marine protected areas: the 15.4 km$^2$ Hol Chan Marine Reserve established in 1987 and the 39.1 km$^2$ Caye Caulker Marine Reserve established in 1998 (http://protectedareas.gov.bz/marine-reserves/). The heavily protected reef portions of these reserves are no-take, no-touch, and they both include ranger surveilance. Visitors are required to be accompanied by a licensed Belize tour guide employed by a licensed tour operator for water entry. The operators and guides at Hol Chan are required to attend annual training on coral reef etiquette and best practices, provided by the reserve. Theoretically, Coral Gardens may benefit from the protection of these adjacent coral reef habitats, as well as the training provided to guides who frequent the area. Herbivore control of macroalgal abundance is robust, reef habitat is conserved, and coral abundance is monitored at these marine reserves, so spawning and recruitment spillover effects could be benefitting Coral

Gardens. Additionally, concentrated tourism at these nearby locations promotes decreased pressure for Coral Gardens.

This study has documented that *Acropora cervicornis* is *persistent* at Coral Gardens. Whether *A. cervicornis* remains *resilient* at this refugium is less certain. Coral Gardens continued to satisfy most criteria for coral reef refugia reviewed by Kavousi and Keppel [25, and references therein]. Long-term buffering (i.e., favorable environmental conditions persisting for decades to centuries) is key to refugium status, and Coral Gardens meets this criterion. *A. cervicornis* has been present at Coral Gardens for more than several decades [16, 18] and has persisted despite a decline in live coral cover in the present study period. It likely meets the accessibility criterium (i.e., target taxon can easily reach the site), as the site may benefit from connectivity with two protected areas.

Coral Gardens is the largest documented extant site for *A. cervicornis* in the Caribbean [27] that we know of, and larger sites support more adjacent species, larger populations, and greater genetic diversity [25]. This study documents that Coral Gardens also satisfies the important criterium of low exposure to other disturbances (i.e., stressors not related to climate change). It is more difficult to assess whether the site meets the criterium of protection from multiple climatic stressors, particularly warming ocean temperatures, intensified storms, and the increasing threat of ocean acidification. Our data suggest *A. cervicornis* at Coral Gardens has been persistent up to the present. What we cannot predict is how resilient this coral will remain in the face of future climate change, as temperature change at this site appears to be the most likely cause for live coral decline during our study period.

## The future of coral gardens

Based on our assessment of potential stressors, we suggest that temperature range, annual mean, and duration of above-average daily temperatures most likely pose the greatest threat to Coral Gardens, assuming status quo for local development factors. Nonetheless, there are many assumptions embedded in this projected future and any of the known stressors to coral reefs that are not currently an apparent threat to Coral Gardens could be activated. Poorly managed coastal development and infrastructure projects could result in increased sedimentation, pollution, or even nutrient loading, although this is less likely since Coral Gardens is not near potential agricultural sites. While herbivores such as urchins and damselfish are protected in Belize, loss of herbivores to natural causes could result in enhanced macroalgal abundance. Sewage pollution from nearby San Pedro could result in eutrophication and ocean acidification poses an increasing potential threat to reefs and could affect Coral Gardens in the future. Most importantly, an increase in storm frequency or intensity coupled with other climate change factors could threaten an environment brought to the brink by global and regional warming and/or temperature variability. Our results emphasize the importance of continued monitoring at this important site. Managing local stressors at the systems level may help mitigate, delay, or buffer negative change, which highlights the importance of the Northern Belize Coastal Complex in managing marine protected areas and reef connectivity in this region.

## Conclusions

*Acropora cervicornis* coral continues to persist at Coral Gardens despite dramatic losses elsewhere in the Caribbean prior to our study. Our data suggest that Coral Gardens remains a refugium for the species for now, but significant declines during the study period, particularly between 2016 and 2017, threatens this designation. The lack of local stressors may have been key in allowing this exceptional *A. cervicornis* coral reef to thrive, and we found no clear correlation between historical data of storm events and *A. cervicornis* mortality at this site. Although

refugia can be an important lifeline for coral species in decline, and may persist for hundreds of years, these habitats are not immune to the effects of *future* climate change. Our data indicate that even in the absence of most reef threats, rising and more variable SSTs may threaten the refugium status of Coral Gardens. We suggest that thermal stress due to early onset, higher than normal temperatures, and sustained heat in 2016 may have made *A. cervicornis* at Coral Gardens more vulnerable to other stressors, including the passage of Hurricane Earl, than would have been the case in prior years. With temperatures rising and storms predicted to increase in frequency and/or duration, we suggest that the future of Coral Gardens as a refugium is uncertain. It seems critical that local stressors remain diminished at this site, but this will not necessarily guarantee sustained live coral abundance. The fate of Coral Gardens, and perhaps similar refugia, will remain increasingly uncertain if storms become more common in concert with rising and more variable temperatures in a warming world.

## Supporting information

**S1 Table. pH measurements in proximity to Coral Gardens collected in 2019.**
(PDF)

**S2 Table. Live coral data from Coral Gardens from 2012–2019.**
(PDF)

**S3 Table. Satellite and sensor sea surface temperature data for Coral Gardens.** Satellite data were obtained as 0.5-deg gridded SST data (*CoralTemp v3.1*) from the Coral Reef Watch 5-km product suite (Heron et al., 2015). Maximum monthly means (MMM) SST were calculated using the gridded data for 1985–1990 + 1993 and DHW were calculated as daily values that exceeded MMM + 1˚C over a 12-week period.
(PDF)

**S4 Table. Sediment thickness data from Coral Gardens from 2014.**
(PDF)

**S5 Table. Coral canopy height data from transects 1 and 5 at Coral Gardens in 2014.**
(PDF)

**S6 Table. Urchin abundance data (n per quadrat) at Coral Gardens in 2014 and 2019.**
(PDF)

**S7 Table. Damselfish abundance data (n per quadrat) at Coral Gardens in 2014 and 2019.**
(PDF)

**S8 Table. Substrate point count data at Coral Gardens in 2014 and 2019.**
(PDF)

**S9 Table. $^{230}$Th analytical age data from Coral Gardens from 1982–2012.**
(PDF)

## Acknowledgments

We thank E. Falls, N. Valdez, K. Mattes, M. Gannon, the Hol Chan Marine Reserve, and F. Cruz at Belize Fisheries Department for logistical and analytical support, as well as the reviewers of this manuscript. We also thank the following students for field and laboratory support:

N An, S Bender, W Benson, J Busch, C Caterham, H. Culbertson, E Elium, G Garcia Greco, M Horne, A Irwin, A Mabaka, Z Martin, D Norvell, E Peeling, W Riley, A Stier, J Villegas, T Waggoner, S Walters, R Waters, MF White, P Zuñiga.

## Author Contributions

**Conceptualization:** Lisa Greer, H. Allen Curran, Karl Wirth.

**Data curation:** Lisa Greer, Ginny Johnson, Lauren McManus, Candice Stefanic.

**Formal analysis:** Lisa Greer, H. Allen Curran, Karl Wirth, Robert Humston, Ginny Johnson, Lauren McManus, Candice Stefanic, Tara Clark, Halard Lescinsky.

**Funding acquisition:** Lisa Greer, H. Allen Curran, Karl Wirth, Ginny Johnson, Lauren McManus, Candice Stefanic, Halard Lescinsky.

**Investigation:** Lisa Greer, H. Allen Curran, Karl Wirth, Robert Humston, Ginny Johnson, Lauren McManus, Candice Stefanic, Tara Clark, Halard Lescinsky, Kirah Forman-Castillo.

**Methodology:** Lisa Greer, H. Allen Curran, Karl Wirth, Tara Clark, Halard Lescinsky.

**Project administration:** Lisa Greer, Karl Wirth.

**Resources:** Lisa Greer, Kirah Forman-Castillo.

**Software:** Lisa Greer.

**Supervision:** Lisa Greer, H. Allen Curran, Karl Wirth.

**Validation:** Lisa Greer, H. Allen Curran, Karl Wirth, Robert Humston, Kirah Forman-Castillo.

**Visualization:** Lisa Greer, Karl Wirth, Robert Humston, Ginny Johnson, Lauren McManus, Candice Stefanic, Tara Clark.

**Writing – original draft:** Lisa Greer.

**Writing – review & editing:** Lisa Greer, H. Allen Curran, Karl Wirth, Robert Humston, Ginny Johnson, Lauren McManus, Candice Stefanic, Tara Clark, Halard Lescinsky, Kirah Forman-Castillo.

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
