## [Decision Letter · Decision Letter 0]

18 Oct 2022

PONE-D-22-24875Coral Gardens Reef, Belize: An Acropora spp. refugium under threat in a warming worldPLOS ONE

Dear Dr. Greer,

Thank you for submitting your manuscript to PLOS ONE. After careful consideration, we feel that it has merit but does not fully meet PLOS ONE’s publication criteria as it currently stands. Therefore, we invite you to submit a revised version of the manuscript that addresses the points raised during the review process.

Please submit your revised manuscript within 90 days. If you will need more time than this to complete your revisions, please reply to this message or contact the journal office at plosone@plos.org. Please include the following items when submitting your revised manuscript:A rebuttal letter that responds to each point raised by the academic editor and reviewer(s). You should upload this letter as a separate file labeled 'Response to Reviewers'.A marked-up copy of your manuscript that highlights changes made to the original version. You should upload this as a separate file labeled 'Revised Manuscript with Track Changes'.An unmarked version of your revised paper without tracked changes. You should upload this as a separate file labeled 'Manuscript'.

We look forward to receiving your revised manuscript.

Kind regards,

James R. Guest, Ph.D.

Academic Editor

PLOS ONE

Journal Requirements:

"This work was supported by the National Science Foundation and the Keck Geology Consortium under Grant No. NSF-REU #1358987 to LG, KW and HL and Grant No. NSF-REU #1659322 to LG and KW. Support also was received from The Washington and Lee University: Johnson Opportunity Grant, Summer Research Scholar Program, Lenfest summer research grant, R. Preston Hawkins IV Award in Geology, and Department of Geology to LG, GJ, LM, CS."

"This work was supported by the National Science Foundation and the Keck Geology Consortium under Grant No. NSF-REU #1358987 to LG, KW and HL and Grant No. NSF-REU #1659322  to LG and KW (https://keckgeology.org/). Support also was received from The Washington and Lee University: Johnson Opportunity Grant, Summer Research Scholar Program, Lenfest summer research grant, R. Preston Hawkinas IV Award in Geology, and Department of Geology to LG, GJ, LM, CS, and ARC DECRA Fellowship support for TC (https://www.arc.gov.au/grants/

discovery-program/discovery-early-career-researcher-award-decra). The funders had no role in study design, data collection and analysis, decision to publish, or preparation of the manuscript."

3. We noted in your submission details that a portion of your manuscript may have been presented or published elsewhere. 

"We do call upon previously aged 230Th data presented in Greer et al. (2020) but we apply the data uniquely to this manuscript."

6. We note that Figures 2 and 7 in your submission contain copyrighted images. All PLOS content is published under the Creative Commons Attribution License (CC BY 4.0), which means that the manuscript, images, and Supporting Information files will be freely available online, and any third party is permitted to access, download, copy, distribute, and use these materials in any way, even commercially, with proper attribution. For more information, see our copyright guidelines: http://journals.plos.org/plosone/s/licenses-and-copyright.

a. You may seek permission from the original copyright holder of Figures 2 and 7 to publish the content specifically under the CC BY 4.0 license. 

Additional Editor Comments:

I received two sets of reviews that identify some issues with the manuscript. Both reviewers agreed that the manuscript was of great value, however one reviewer did notice an issue with the calculation of DHWs. I agree with them on this point and suggest that you look at using a more relevant climatology for calculating MMM. This may change some of your results and interpretation, so I've suggested major revision.

Reviewers' comments:

Reviewer's Responses to Questions

**Comments to the Author**

1. Is the manuscript technically sound, and do the data support the conclusions?

Reviewer #1: Yes

2. Has the statistical analysis been performed appropriately and rigorously? 

Reviewer #1: No

3. Have the authors made all data underlying the findings in their manuscript fully available?

Reviewer #1: No

4. Is the manuscript presented in an intelligible fashion and written in standard English?

Reviewer #1: Yes

5. Review Comments to the Author

Reviewer #1: This manuscript describes an assessment of threats to one of the largest remaining healthy Acropora cervicornis patches in the Caribbean. Previous work by the lead author has identified this field site as a refugium of Acropora spp. and this paper expands that work with many years of intensive field monitoring and a variety of types of ancillary data to analyze the potential stability of the refugium. The manuscript is clearly written and generally a valuable contribution to the field. It is important for future conservation of endangered species to recognize refugia and recognize the potential threats that could destroy them and the species they protect.

In my review submission, I indicate that the data are not available because I could not find a supplement and the data were not all in the manuscript. There were summary tables in the text, but not all of the raw data. I would have expected a table of daily and monthly temperature data, as well as all of the live coral abundance data for each transect, the pH data, the substrate composition data, sediment size and composition data, canopy height, sediment thickness and depth to hardground, urchin counts, and damselfish survey data.

I have one main concern with the analysis as it is written. The authors use the concept of degree heating weeks, which has been used for many years to quantify coral bleaching risk. However, the calculations described in the present text indicate incorrect application of the concept, according to my understanding. The issue is that DHW is supposed to be relative to the mean maximum monthly temperature (MMM) during a base period when bleaching was not common. Using the monthly data for each year during the study period, 2012-2020, implicitly assumes instantaneous adaptation of coral populations to current warming trends, which is a bad assumption. NOAA coral reef watch, which has popularized this metric uses a base period of 1985–1990+1993 for historic reasons (see Heron et al 2015). I recommend using the same base period.

I understand that a problem with the 1980s-1990s base period is that you don’t have local in-situ data as you do for the 2012-2020 period. However global gridded SST products are widely available and often track local reef temperatures fairly well (Winter et al, 1998). A calibration of the monthly in-situ data with gridded data during the period of common data (as in Smith et al., 2006) will permit you to adjust the available gridded temperature to better match the expected larger variance of a local record during the earlier base period. I think this is will be a much better estimate of MMM against which to calculate DHW.

Another potential issue with the DHW concept as applied is that recent work (Lachs et al, 2021) indicates that a lower threshold of MMM+0 for 8 weeks duration may better capture thermal stress in corals. I am ambivalent about the need to use this lowered threshold as I agree with Lachs et al., 2021 that spatial variations in coral response are likely significant. However, I think you will find that by using a more reasonable base period for the climatological MMM, you will have substantially more accumulation of DHW during the course of the study, especially 2016.

Some specific comments for small corrections and a list of the references cited above follow.

SPECIFIC COMMENTS

Line 117 The description “high resolution” is very unclear for multiple reasons. Is the resolution over space or time, and how high is “high”? The compound adverb (which needs to be written “high-resolution” to be correct) describes the calculations, but I don’t think the calculations had high resolution, instead I think the spatial data used for the calculations were of cm-scale resolution. To a microscopy specialist, cm-scale is exceedingly low resolution. Further the syntax using “high-resolution” first makes a reader wonder if all the data in the list is “high-resolution”, or if the authors are only referring to the calculations of live coral cover.

Line 119. I would argue that all of the data in this list are observational data. Perhaps the authors mean “personal observations” of conditions at the site over the years.

Figure 1: This figure would be easier to see as a two-panel figure. The inset map is very small and highly detailed, yet the base map has little fine detail and is rather large. They would both be more appropriately sized if they were equal and side-by-side.

Figure 2: the caption describes a m2 quadrat, but I think this would be better described as a 1 m2 quadrat. Additionally, the caption describes the coral image traced with Adobe Photoshop, but in the text, Adobe Illustrator is attributed. This should be consistent.

REFERENCES

Heron, S.F.; Liu, G.; Eakin, C.M.; Skirving, W.J.; Muller-Karger, F.E.; Vera-Rodriguez, M.; de la Cour, J.L.; Burgess, T.F.R.; Strong, A.E. ; Geiger, E.F.; et al. Climatology Development for NOAA Coral Reef Watch’s 5-km Product Suite, NOAA Technical Report; NESDIS: Silver Spring, MD, USA, 2015; Volume 145.

Lachs, L.; Bythell, J.C; East, H.K; Edwards, A.J; Mumby, P.J; Skirving, W.J; Spady, B.L; Guest, J.R. Fine-Tuning Heat Stress Algorithms to Optimise Global Predictions of Mass Coral Bleaching. Remote Sens. 2021, 13, 2677. https://doi.org/10.3390/rs13142677

Smith, J. M., Quinn, T. M., Helmle, K. P., and Halley, R. B. (2006), Reproducibility of geochemical and climatic signals in the Atlantic coral Montastraea faveolata, Paleoceanography, 21, PA1010, doi:10.1029/2005PA001187.

A. Winter, R. S. Appeldoorn, A. Bruckner, E. H. Williams. Jr., C. Goenaga, Sea surface temperatures and coral reef bleaching off La Parguera, Puerto Rico (northeastern Caribbean Sea). Coral Reefs (1998) 17 : 377-382.

6. PLOS authors have the option to publish the peer review history of their article (what does this mean?). If published, this will include your full peer review and any attached files.

Reviewer #1: No

Reviewer #2:

See attached PDF of paper - see comments in drop down tabs

Summary

1.  accept manuscript with revision

2.  limitation of study - one field visit (snapshot) per year - hard to infer causality for the intervening 11 months  Need to discuss this limitation

3.  No repeated measures of coral colonies during period of loss (mortality) -

4.  Can only guess as to the cause of mortality ???

5.  Agree that temperature is likely proximal driver of ACER loss

6.  Personal observation to my one visit to site = There is also a minor level of background WBD at Coral Gardens (~1%) - this could have jumped as temperatures reached their maximum in late summer /early fall causing a localized but temporally limited mortality event.

see the attached references

Randall, C.J. and van Woesik, R., 2015. Contemporary white-band disease in Caribbean corals driven by climate change. *Nature Climate Change*, *5*(4), pp.375-379.

Gignoux-Wolfsohn, S.A., Precht, W.F., Peters, E.C., Gintert, B.E. and Kaufman, L.S., 2020. Ecology, histopathology, and microbial ecology of a white-band disease outbreak in the threatened staghorn coral Acropora cervicornis. *Diseases of Aquatic Organisms*, *137*(3), pp.217-237.

7.  Just as easy to infer that disease/bleaching was as important as passage of hurricane

8.  Hurricane wave stress plus temperature stress may have led to shut down reaction (speculation on my part - but possible)

see SDR

Antonius, A., 1977. Coral mortality in reef: a problem for science and management. In: Proc. 3rd Int. Coral. Reef. Symp, 2, pp. 618–623.

9.  Manuscript is well written, stats ok, good use of references

10.  Broad implication for linking of climate change science with coral loss

---

## [Author Response · Author response to Decision Letter 0]

4 Dec 2022

We are grateful four the thoughtful reviews and address all issues in the Response to Reviewers document here submitted.

---

## [Editor Report · Decision Letter 1]

10 Jan 2023

Coral Gardens Reef, Belize: An Acropora spp. refugium under threat in a warming world

PONE-D-22-24875R1

Dear Dr. Greer,

We’re pleased to inform you that your manuscript has been judged scientifically suitable for publication and will be formally accepted for publication once it meets all outstanding technical requirements.

Kind regards,

James R. Guest, Ph.D.

Academic Editor

PLOS ONE
---

## [Editor Report · Acceptance letter]

30 Jan 2023

PONE-D-22-24875R1 

Coral Gardens Reef, Belize: An *Acropora* spp. refugium under threat in a warming world 

Dear Dr. Greer:

I'm pleased to inform you that your manuscript has been deemed suitable for publication in PLOS ONE. Congratulations! Your manuscript is now with our production department. 

Kind regards, 

on behalf of

Dr. James R. Guest 

Academic Editor

PLOS ONE